# Measures of Oxidative Status Markers in Relation to Age, Sex, and Season in Sick and Healthy Captive Asian Elephants in Thailand

**DOI:** 10.3390/ani13091548

**Published:** 2023-05-05

**Authors:** Worapong Kosaruk, Janine L. Brown, Patcharapa Towiboon, Veerasak Punyapornwithaya, Kidsadagon Pringproa, Chatchote Thitaram

**Affiliations:** 1Doctoral Degree Program in Veterinary Science, Faculty of Veterinary Medicine, Chiang Mai University, Chiang Mai 50100, Thailand; 2Center of Elephant and Wildlife Health, Chiang Mai University Animal Hospital, Chiang Mai 50100, Thailand; 3Elephant, Wildlife, and Companion Animals Research Group, Chiang Mai University, Chiang Mai 50100, Thailand; 4Center for Species Survival, Smithsonian Conservation Biology Institute, Front Royal, VA 22630, USA; 5Department of Food Animal Clinic, Faculty of Veterinary Medicine, Chiang Mai University, Chiang Mai 50100, Thailand; 6Department of Veterinary Bioscience and Veterinary Public Health, Faculty of Veterinary Medicine, Chiang Mai University, Chiang Mai 50100, Thailand; 7Department of Companion Animal and Wildlife Clinic, Faculty of Veterinary Medicine, Chiang Mai University, Chiang Mai 50100, Thailand

**Keywords:** Asian elephants, oxidative stress, reactive oxygen species, malondialdehyde, albumin, glutathione peroxidase, catalase

## Abstract

**Simple Summary:**

Oxidative stress is an adverse condition in animals that can cause tissue damage and result in poor health. A number of factors can contribute to this condition, although little has been studied empirically in elephants. This study measured multiple serum oxidative status markers and a stress biomarker (fecal glucocorticoid metabolites) to examine the effects of age, sex, sampling season, and disease factors in captive Asian elephants in Thailand. Age and season influenced several biomarkers and should be considered in data interpretation, while sex did not. The most significant changes in oxidative and antioxidative activity were associated with elephant endotheliotropic herpesvirus hemorrhagic disease, a highly fatal disease in calves, suggesting a strong link with oxidative stress responses.

**Abstract:**

Oxidative stress is a pathological condition that can have adverse effects on animal health, although little research has been conducted on wildlife species. In this study, blood was collected from captive Asian elephants for the assessment of five serum oxidative status markers (reactive oxygen species (ROS) concentrations; malondialdehyde, MDA; albumin; glutathione peroxidase, GPx; and catalase) in healthy (*n* = 137) and sick (*n* = 20) animals. Health problems consisted of weakness, puncture wounds, gastrointestinal distress, eye and musculoskeletal problems, and elephant endotheliotropic herpesvirus hemorrhagic disease (EEHV-HD). Fecal samples were also collected to assess glucocorticoid metabolites (fGCMs) as a measure of stress. All data were analyzed in relation to age, sex, sampling season, and their interactions using generalized linear models, and a correlation matrix was constructed. ROS and serum albumin concentrations exhibited the highest concentrations in aged elephants (>45 years). No sex differences were found for any biomarker. Interactions were observed for age groups and seasons for ROS and catalase, while GPx displayed a significant interaction between sex and season. In pairwise comparisons, significant increases in ROS and catalase were observed in summer, with higher ROS concentrations observed only in the adult female group. Lower catalase activity was exhibited in juvenile males, subadult males, adult females, and aged females compared to subadult and adult elephants (males and females) in winter and the rainy season. There was a positive association between catalase activity and fGCMs (r = 0.23, *p* < 0.05), and a number of red blood cell parameters were positively associated with several of these biomarkers, suggesting high oxidative and antioxidative activity covary in red cells (*p* < 0.05). According to health status, elephants with EEHV-HD showed the most significant changes in oxidative stress markers, with MDA, GPx, and catalase being higher and albumin being lower than in healthy elephants. This study provides an analysis of understudied health biomarkers in Asian elephants, which can be used as additional tools for assessing the health condition of this species and suggests age and season may be important factors in data interpretation.

## 1. Introduction

The Asian elephant (*Elephas maximus*) is listed as endangered in the International Union for Conservation of Nature Red List (IUCN, 2015), with declining numbers throughout their existing ranges [1]. Thailand currently has a wild population of about 3200 individuals [2], with upwards of 3800 elephants in captivity [3] that play an important role in wildlife-based ecotourism [4]. While wild numbers have been fairly stable for the past decade, the captive population is not self-sustaining due to fewer births than deaths [5]. Captive elephants in Asia experience a number of disease conditions, such as colic, trauma, poor body condition, and infections [6], with some elephants appearing to be more susceptible than others. For example, only 20% of elephant calves infected with elephant endotheliotropic herpesvirus (EEHV) show severe clinical signs, while the other 80% are asymptomatic or have only mild symptoms [7]. In addition, similar to the human herpesvirus, EEHV appears to be a latent infection among elephant herds [8], with a lack of changes in standard blood tests or clinical signs during the latency stage causing challenges in disease prevention and control [7]. Similar to other species, the primary diagnostic tool for assessing sick elephants involves blood analyses (e.g., complete blood counts and serum chemistries) to identify potential causes and plan treatment options [9,10]. However, not all conditions result in clear hematological changes, such as those associated with oxidative stress, which can contribute to disease progression and limit treatment success [11,12]. Thus, we need additional tools to better understand the pathophysiology of elephant disease, which could lead to more targeted diagnostics and treatments.

Free radicals are natural products of oxygen metabolism in living creatures, mainly in the form of reactive oxygen species (ROS) that have highly reactive properties [13]. Normally, low concentrations of free radicals play an important role in normal cell homeostasis, assisting the modulation of various cytokines and growth factors and being involved in the intracellular apoptosis process [14,15]. Pathological conditions, such as aging, trauma, cancer, and disease, can cause the overproduction of ROS, leading to an imbalance between free radicals and antioxidants, a condition known as ‘oxidative stress’ [16]. If the body cannot rid itself of excess free radicals through the action of antioxidants, damage to macrobiomolecules (lipids, proteins, and DNA/RNA) ensues, triggering cell death [14,16]. Several studies have shown that oxidative stress is associated with pathological conditions that can lead to disease advancement and reduced health fitness [11,12,17]. There are a number of factors that induce oxidative stress, including poor nutrition, environmental effects (i.e., seasonal changes), and diseases [18,19]. Increased glucocorticoid concentrations can also be an underlying mechanism behind increased oxidative stress, subsequently contributing to disease processes [20]. However, the impact of oxidative stress on health may not always be obvious in terms of clinical signs and external appearances [12,21]. Thus, measures of oxidative stress markers could provide useful information to identify animals at risk of poor health and welfare outcomes [15,22].

Several serum biomarkers have been used to document oxidative status in animals and humans [23,24]. Oxidation markers are those modified by interactions with ROS and involved in oxidative stress conditions, while antioxidant markers are the body’s defense mechanism against oxidative damage [25]. Methods to measure total ROS concentrations have been developed for many species [13,21], providing a means to assess redox status and associations with health outcomes. As an example, an increase in ROS production can lead to herpesvirus replication and reactivation in humans and horses because it creates a proper environment for the virus to replicate [12,26]. Another frequently used oxidative marker is malondialdehyde (MDA), which is a product of the lipid peroxidation process. It is a highly toxic molecule that causes damage by the destruction of cell membranes [27]. Several studies have reported significantly higher MDA concentrations in various metabolic and cardiovascular diseases [28,29].

Antioxidant markers include albumin and enzymes, such as glutathione peroxidase (GPx) and catalase. Serum albumin is the most abundant protein in mammalian serum, serving many physiological functions, including as an important extracellular antioxidant through its ROS-binding properties [30]. It is the main defense mechanism for preventing oxidative damage by scavenging highly reactive molecules [31]. GPx and catalase play major roles in converting hydrogen peroxide (a major contributor to oxidative damage) to water, thus preventing the production of unstable molecules that can cause cellular damage [32,33]. Decreased activity of antioxidant enzymes has been linked to health problems in humans and animals, such as tumor development [34] and metabolic disorders [35,36]. High oxidative and low antioxidant activity has also been linked to infectious hemorrhagic-related diseases in humans (e.g., leptospirosis, dengue, and malaria infections) [37,38,39] and generally to increased bodily function impairments, disease progression, and mortality [40,41]. Hence, similar mechanisms may be at play in elephants in relation to severe hemorrhagic conditions, such as that caused by EEHV.

Measuring biomarkers associated with oxidative status could be beneficial to better understand the pathophysiology of various health conditions in elephants, as well as assess the efficacy of tools for disease monitoring and treatment. Thus, the objective of this study was to analyze five oxidative status markers (MDA, ROS, albumin, GPx, and catalase), in addition to fecal glucocorticoid metabolites (fGCMs) to examine the differences between healthy and sick captive Asian elephants in Thailand, as well as in relation to age, sex, and season.

## 2. Materials and Methods

### 2.1. Elephants and Sample Collection

This study was approved by the Faculty of Veterinary Medicine (FVM), Chiang Mai University (CMU), Animal Care and Use Committee (Number S7/2564).

A total of 137 healthy Asian elephants with no clinical signs during sample collection, as well as 1 month before and after sampling, participated in this study (mean age, 16.8 ± 16.4 years; range, 1–61; 45 males, 92 females). Elephants were housed at tourist camps throughout Thailand, mostly in the north (Chiang Mai and Lampang provinces), as shown in Table 1. Elephants were categorized into four age groups based on Angkawanish et al. [6]: (1) juvenile, 1–5 years (*n* = 45; mean, 3.1 ± 1.2 years); (2) subadult, 6–10 years (*n* = 36; mean, 8.0 ± 1.1 years); (3) adult, 11–45 years (*n* = 45; mean, 29.3 ± 10.5 years); and (4) aged, >45 years (*n* = 11; mean, 50.9 ± 5.8 years).

Another 20 elephants (mean age, 27.6 ± 20.0 years; 4 males, 16 females) that presented with weakness (*n* = 1), puncture wounds (*n* = 2), gastrointestinal (GI) distress (*n* = 5), eye problems (*n* = 3), musculoskeletal (MS) conditions (*n* = 4), and elephant endotheliotropic herpesvirus hemorrhagic disease (EEHV-HD) (*n* = 5) were also evaluated to compare with healthy elephants. The ‘weakness’ elephant had gradually lost weight over a 6-month period and been unable to stand after laying down, with death occurring 7 days after sample collection; cause of death was unknown. Both elephants with puncture wounds had developed subcutaneous abscesses that were present for >1 month. For GI cases, elephants exhibited hypomotility (<2 days) and were unable to pass feces leading to acute abdominal pain. Eye problems involved corneal ulcer and opacity with a runny eye. For the MS group, two elephants presented with broken bones (one foreleg fracture and one hindleg fracture), while the other two exhibited muscle pain and lameness. For EEHV-HD, calves had acute illness with anorexia, high fever, and facial edema; all were confirmed by real-time polymerase chain reaction analysis of whole blood (Veterinary Diagnostic Laboratory, CMU). Sick elephants were diagnosed by and under the care of a mobile clinic organized by the CMU Animal Hospital, Center of Elephant and Wildlife Health or elephant hospital (Thailand Elephant Conservation Center, Lampang province). All elephants were fed mainly fresh roughage (e.g., grasses, corn stalks, and pineapple leaves) with high-energy fruits (e.g., bananas, sugar cane) provided occasionally.

Blood samples were collected one time from each elephant during different seasons of the year from an ear vein using standard phlebotomy and placed into EDTA (0.5 mL, BD Vacutainer^®^ K2 EDTA 5.4 mg, Franklin Lakes, NJ, USA) and red-top serum (5 mL, BD Vacutainer^®^ Serum, Franklin Lakes, NJ, USA) tubes. For collection of serum, blood was allowed to clot at room temperature (RT) for 1 h, transported in a cool box to the laboratory at the FVM-CMU, and centrifuged at 700× *g* (Hettich, Germany) for 10 min. Serum was stored at −80 °C for up to 3 months until oxidative stress marker analyses. Serum blood chemistry (blood urea nitrogen, BUN; creatinine; alanine transaminase, ALT; alkaline phosphatase, ALP; total protein) was analyzed using automated clinical chemistry analyzer (BX-3010, Sysmex Asia Pacific Pte Ltd., Singapore). EDTA blood was chilled (4 °C) and used for complete blood counts conducted immediately after transportation to FVM-CMU (within 6–12 h) by auto hematology analyzer (BC-5300, Mindray™, Shenzhen, China), whereas white blood cell differential counts were performed manually on Wright–Giemsa-stained blood smears, similar to sample processing protocols developed for humans [42]. Fresh fecal samples (~20 g) were also collected from natural defecation (*n* = 96), put in a zip-lock bag, transported in a cool box to the laboratory, and stored frozen at −20 °C until processing and analysis.

### 2.2. Serum Oxidative Status Markers

ROS concentrations were assessed according to Hayashi et al. [13] with some modifications. In brief, sodium acetate buffer (140 µL, 0.1 M, pH 4.8) was added to 96-well microplates (Nunc™, F96 Maxisorp immune plate, Roskilde, Denmark) followed by undiluted serum or hydrogen peroxide (H_2_O_2_) standard solution (25 µL, 0 to 250 mg/L, Merck, Germany) and incubated for 5 min at RT. Then, 100 µL of a mixture of N, N-diethyl-para-phenylenediamine (DEPPD; 500 µg/mL, Sigma-Aldrich, St. Louis, MO, USA) and 21.85 mM ferrous sulfate (Loba Chemie PVT. LTD., Mumbai, India) (1:5, *v*/*v*) was added to each well. The absorbance was read at 492 nm every minute for up to 6 min. Concentrations of ROS were calculated from the linear slope of the calibration curve (absorbance increase at 492 nm/min × 1000) and expressed as units of H_2_O_2_ (1 unit = 1.0 mg H_2_O_2_/L). Assay sensitivity was 1.613 mg/L.

MDA concentrations were quantified using the thiobarbituric acid reactive substances (TBARSs) assay described by Satitmanwiwat et al. [43]. In brief, samples or standards (1.25 to 80 µL) (50 µL) were mixed in a test tube with 750 µL of phosphoric acid (0.44 M, H_3_PO_4_), 250 µL of thiobarbituric acid (42 nM), and 450 µL of distilled water. Tubes were placed into a boiling water bath for 15 min, cooled on ice for 5 min, and then centrifuged at 900× *g* for 5 min. Absorbance of the supernatant was measured at 532 nm using a UV-VIS spectrophotometer (Shimadzu, Japan). MDA concentrations were calculated from standard curves of MDA equivalents generated by the acid-catalyzed hydrolysis of 1,1,3,3-tetramethoxypropane (TMP) (5–80 µL). Concentrations of MDA were expressed as nmol/mL. 

Serum albumin was quantified using an automated chemistry analyzer (BX-3010, Sysmex Corporation, Tokyo, Japan) according to the manufacturer protocol.

GPx activity was analyzed based on Ahmed et al. [44] with some modifications. In brief, serum samples (12 µL) and reduced glutathione (25 µL, 4 nM, Sigma-Aldrich, St. Louis, MO, USA) were incubated in a phosphate buffer solution (50 mM of KH_2_PO_4_, pH 7.0). The reaction was initiated by adding peroxide solution (H_2_O_2_, 25 µL), vortexing thoroughly, and incubating at 37 °C for 10 min. Trichloroacetic acid (8%, 62 µL, RCI Labscan, Bangkok, Thailand) was added to stop the reaction, followed by vortexing and centrifugation at 700× *g* for 10 min. Supernatants were pipetted into 96-well microplates in duplicate (50 µL/well) (TECAN, Männedorf, Switzerland) followed by freshly prepared CUPRAC reagent (150 µL; 10mM CuCl_2_, 1.816 M NH_4_Ac, and 7.5 mM Nc) at a ratio of 1:1:1 (*v*/*v*/*v*). Absorbance was read at 450 nm against a blank, and the GPx activity expressed as U/L according to Ahmed et al. [44]. Assay sensitivity was 0.023 U/L.

Catalase activity was measured using the spectrophotometric method of Hadwan and Ali [45] with some modifications. In brief, catalase from bovine liver (Sigma-Aldrich, USA) was used as the standard (0.5–10 U/mL) and high- and low-quality control samples. Serum (50 µL) and catalase standard (50 µL) were added to a 96-well microplate in duplicate, and H_2_O_2_ solution (100 µL, 10 mM) was added. Plates were incubated at 37 °C for 5 min, after which ammonium metavanadate reagent (100 µL, 10 mM NH_4_VO_3_ in 50 mM H_2_SO_4_ solution) was added, and the plates were incubated again at 25 °C for 10 min before reading the absorbance at 450 nm. Catalase activity was expressed as U/mL. Phosphate buffer solution (150 µL, 50 mM of KH_2_PO_4_, pH 7.0) was used as a blank. Recovery of serum catalase activity added to a low-concentration sample before analysis was significant (y = 1.0447x + 0.9974, R^2^ = 0.9992). Assay sensitivity was 3.282 U/mL. The intra- and interassay coefficients of variation were less than 10%.

### 2.3. Fecal Glucocorticoid Metabolites

Fecal samples were extracted following the protocol of Kosaruk et al. [46]. Briefly, frozen samples were thawed at RT before drying in a conventional oven at 60 °C for 24–48 h. Dried fecal powder (0.1 g ± 0.001 g) was placed into glass tubes, and 5 mL of 90% ethanol (EtOH):10% distilled water was added, followed by vortexing briefly before placing in a water bath (90 °C) to boil for 20 min. During this step, EtOH was occasionally added to maintain the volume at 5 mL. After boiling, the tubes were centrifuged at 960× *g* for 20 min, and the supernatant was poured into a new tube. The pellet was extracted again, and the supernatants combined. Samples were dried in a 90 °C water bath followed by resuspension with 3 mL of 95% EtOH before drying down again. Final fecal extracts were resuspended in 1 mL of 50% methanol and stored at −20 °C until analysis.

Fecal extracts were diluted 1:3 in assay buffer for quantification of fGCM concentrations using a double-antibody corticosterone enzyme immunoassay [46]. Samples and corticosterone standard (50 µL) were added in duplicate to a 96-well microplate plate pre-coated with antirabbit IgG, followed by corticosterone-HRP (25 µL; 1:30,000) and anticorticosterone antibody (25 µL; 1:100,000, CJM006, Coralie Munro, University of California, Davis). Plates were incubated at RT for 2 h, and then 100 µL of TMB substrate solution (KPL TMB Microwell Peroxidase Substrate System 2-contents) was added and incubated in the dark for 20 min. Stop solution (50 µL, 1 N HCl) was added, and the absorbance measured at 450 nm. Assay sensitivity was 0.11 ng/mL, and the intra- and interassay coefficients of variation were <10% and 11.3%, respectively.

### 2.4. Statistical Analysis

All data were analyzed using R statistical software (version 4.1.0, RStudio, 2021). The normality of data distribution was examined using quantile-to-quantile plots. Data for oxidative stress markers (ROS, MDA, albumin, GPx, and catalase) are shown as a mean ± standard deviation of the mean (SD), and reference range of all parameters was calculated (‘referenceIntervals’ package). All oxidative status and stress biomarkers were analyzed using generalized linear models (GLM, ‘glm’ package) with age group, sex, and season according to sampling date (summer, 16 February–15 May, average temperature, 29 °C, humidity 56%; rainy, 16 May–15 October, average temperature, 28 °C, humidity 83%; winter, 16 October–15 February, average temperature, 24 °C, humidity 65%) (Thai Meteorological Department, www.tmd.go.th (accessed on 19 January 2023) as the main effects, as well as interactions. Temperature–humidity indexes (THI) for each season were calculated following Yeotikar et al. [47]: summer THI = 78.3; rainy THI = 80.2; winter THI = 72.6. Tukey post hoc tests were further used for pairwise comparisons (‘emmeans’ package). A subset of age- or sex- or camp-matched healthy elephants (*n* = 30) were selected to compare biomarker data with EEHV-HD (*n* = 5) and other illness conditions (*n* = 15) using Kruskal–Wallis tests followed by Tukey post hoc tests. Data on age and sex of sick and selected healthy elephants are shown in Appendix A. Pearson correlation tests were performed to determine relationships between oxidative stress markers, fGCMs, and blood parameters (red blood cells, RBCs; total white blood cells, WBCs; heterophils, HETs; lymphocytes, LYMs; monocytes, MONOs; platelets, PLTs) in all healthy elephants. Statistical significance for all tests was set at *p* < 0.05.

## 3. Results

Box plots and descriptive statistics for oxidant (ROS, MDA) and antioxidant (albumin, GPx, catalase) markers and fGCM concentrations of healthy elephants (*n* = 137) across age groups are shown in Figure 1 and Table 2, respectively. Differences in oxidative status markers between selected healthy (*n* = 30) and sick elephants are shown in Table 3, with box plot data shown in Figure 2. There were no differences in biomarker concentrations between the selected healthy (*n* = 30) and all healthy elephants (*n* = 107) (*p* > 0.05) (Appendix A). Biomarker concentrations across sick elephant categories (*n* = 20) are displayed in Appendix A.

### 3.1. Oxidant Markers

The GLM analysis of serum ROS is shown in Appendix A, with adult and aged groups being significantly different from the reference value (juvenile, *p* < 0.05) (Figure 1). No differences were found for the other main effects (sex and season) (*p* > 0.05). There was a significant interaction between age group and season for adult (2.17 ± 0.11 mg/L, *n* = 33) and aged (2.22 ± 0.08 mg/L, *n* = 9) elephants, with the winter season being different from the reference value (juvenile x summer; 2.18 ± 0.15 mg/L, *n* = 10). Post hoc pairwise comparisons showed that ROS concentrations in female adult elephants in summer (2.35 ± 0.07 mg/L, *n* = 12) were significantly higher than juveniles and subadults in the rainy season for both sexes (juvenile male: 2.10 ± 0.21 mg/L, *n* = 13; juvenile female: 2.09 ± 0.16 mg/L, *n* = 18; subadult male: 2.04 ± 0.12 mg/L, *n* = 7; subadult female: 2.15 ± 0.19 mg/L, *n* = 15) (*p* < 0.05) and were also higher than female adults in winter (2.16 ± 0.11 mg/L, *n* = 25) (*p* < 0.01). However, there were no differences in ROS concentrations between healthy and sick elephants (Table 3 and Figure 2).

The GLM analysis for MDA is presented in Appendix A. No significant differences were found for the main effects (age group, sex, and season) or their interaction or in the post hoc GLM test. According to health status, MDA concentrations in selected healthy elephants were ~30% lower than in sick animals, with concentrations in EEHV-HD elephants being more than double (*p* < 0.01) (Table 3 and Figure 2).

### 3.2. Antioxidant Markers

For serum albumin, the GLM analysis is shown in Appendix A. Aged elephants had significantly higher mean albumin concentrations than the reference value (juvenile, *p* < 0.05) (Figure 1). No significant differences in albumin concentrations were found in relation to the sex or sampling season nor were there any significant interactions (*p* > 0.05). Additionally, no significant differences were found in the post hoc tests. With health status, concentrations in EEHV-HD elephants were only 68% of those of selected healthy elephants (*p* < 0.01) (Table 3 and Figure 2).

The GLM analysis of GPx activity is shown in Appendix A. No significant differences were found across the three main factors (age group, sex, and sampling season), although there was a significant interaction with female elephants in the rainy season (1.28 ± 0.56 U/l, *n* = 33) compared to the reference value (male x summer; 1.10 ± 0.51 U/l, *n* = 12, *p* < 0.05). However, pairwise comparisons found no significant differences with regard to age, sex, or season (*p* > 0.05). Overall mean GPx concentrations did not differ between selected healthy and all sick elephants but were higher in the EEHV-HD group (*p* < 0.05) (Table 3 and Figure 2).

The GLM analysis associated with catalase activity is shown in Appendix A. No significant differences were found among the three main effects (age group, sex, and season) (*p* > 0.05), but there was an interaction for the subadult group in winter, which had significantly higher catalase activity (17.49 ± 7.50 U/mL) when compared to the reference value (juvenile x summer; 10.73 ± 3.90 U/mL, *p* < 0.01). Post hoc pairwise tests showed significant differences, with juvenile (10.24 ± 3.01 U/mL, *n* = 8) and subadult (7.33 ± 3.29 U/mL, *n* = 4) males being lower in summer compared to subadult males in winter (24.54 ± 5.22 U/mL, *n* = 2) (*p* < 0.05). Subadult males in summer also had lower concentrations than adult males in winter (18.46 ± 6.68 U/mL, *n* = 8) (*p* < 0.01). Adult females in summer (7.12 ± 2.97 U/mL, *n* =12) exhibited significantly lower catalase activity than subadult females in the rainy season (14.91 ± 4.52 U/mL, *n* = 15), and subadult males (24.54 ± 5.22 U/mL, *n* = 2), adult males (18.46 ± 6.68 U/mL, *n* = 8), and adult females (15.79 ± 4.82 U/mL, *n* = 25) in winter (*p* < 0.01). Aged females in summer (7.17 ± 0.28 U/mL, *n* = 2) exhibited lower catalase concentrations than the subadult male group in winter (24.54 ± 5.22 U/mL, *n* = 2) (*p* < 0.05). With respect to health status (Table 3 and Figure 2), catalase activity in EEHV-HD-infected elephants was almost double that of selected healthy and other sick elephants (*p* < 0.05).

### 3.3. Fecal Glucocorticoid Metabolites

The GLM analysis of fGCM data is displayed in Appendix A. No significant differences were found in relation to age group, sex, sampling season, or their interactions (*p* > 0.05). Pairwise comparison tests from the GLM model did not find any differences across groups (*p* > 0.05). For health status (Table 3 and Figure 2), there were no differences in fGCM concentrations between healthy and sick elephants (*p* > 0.05), although the variability in sick elephants was much higher based on SDs.

### 3.4. Complete Blood Counts and Serum Chemistry

The complete blood count and serum chemistry data of healthy and sick elephants are displayed in Table 4. Blood parameters of all sick elephants (*n* = 20) in each disease category are displayed in Appendix A. Three red cell parameters (packed cell volume, PCV; hemoglobin; red blood cell count, RBC) were slightly lower in one elephant displaying general weakness (PCV 27%, hemoglobin 9.9 g/dL, RBC 2.23 × 10^6^ cells/µL) compared to the reference range. The mean corpuscular hemoglobin concentrations (MCHCs) reached the minimum limit of the reference range in the EEHV-HD group. For white cells, heterophil counts in EEHV-HD elephants were more than twice as high compared to the healthy and sick groups and outside the reference range. The calculated heterophil-to-lymphocyte (H:L) ratio was also nearly twice as high in EEHV-HD elephants compared to healthy animals; the ratio for other sick elephants was intermediate but still within the range except for the one ‘weakness’ elephant (H:L ratio = 2.0). Monocyte-to-heterophil (M:H) ratios in EEHV-HD elephants were only ~20% of those in healthy and sick animals. Platelet counts in the EEHV-HD group were markedly lower (~30%) than in healthy and sick animals, with the latter two still within the range.

For serum chemistry, creatinine concentrations were over the reference range in the EEHV-HD group and almost twice as high as in healthy and sick elephants. ALT concentrations were well outside the range in the two disease conditions, with the sick and EEHV-HD groups being over 4- and 5-fold higher compared to the healthy group, respectively. The high mean value in the sick group was due to one ‘weakness’ elephant with an ALT concentration of 37 U/L (Appendix A). Similar to platelet counts and albumin concentrations, total serum protein concentrations in EEHV-HD elephants dropped by over a quarter when compared to healthy and sick animals, while other disease conditions were within the reference range. 

### 3.5. Correlation Matrix

The correlations between oxidative status markers, blood differentials, and fGCMs in healthy elephants are shown in Table 5. Weak positive correlations were found between MDA and albumin and GPx concentrations. No other significant correlations among the other oxidative markers were observed. For fGCMs, catalase and platelets showed weak positive correlations, while ROS and RBC were negatively correlated to fGCMs. A number of blood parameters were moderately correlated, particularly among WBC counts.

## 4. Discussion

This is the first study to analyze multiple oxidative status markers, both oxidant and antioxidant, in healthy and sick Asian elephants and examine relationships with serum chemistry and fGCM assessments. Reference ranges for this population were also calculated to begin exploring the potential for using biomarker measures to better understand the factors associated with disease processes. Sex and season had no significant effects on any of the biomarkers, while ROS and albumin increased with age. There were a few interactions, including age by season for ROS and catalase, and sex by season for GPx. For pairwise comparisons of the three main factors (age groups, sex, and seasons), only ROS and catalase were significantly affected. Compared to healthy animals, the sick elephant group (i.e., those displaying weakness, puncture wounds, GI distress, and eye or MS problems) had overall higher concentrations of MDA only, although there were some individuals that exceeded reference range values for other biomarkers. More changes were observed in elephants diagnosed with EEHV-HD, including: (1) higher heterophil counts and H:L ratios, and MDA, catalase, GPx, creatinine, total protein, and ALT concentrations, and (2) lower albumin concentrations, MCHC, and platelet counts. Measures of fGCMs were less informative of disease status, with the exception of a few very sick animals (one weakness, two GI cases) that had high concentrations. These data reveal significant relationships between a number of oxidative stress and health markers with some disease conditions, suggesting that assessing oxidative status in addition to more traditional measures may aid in disease diagnosis and treatment. 

For ROS, the results show that concentrations increased with age, being higher in aged than juvenile elephants. In humans, oxidative stress gradually develops during aging [14,16,48], which can then contribute to many age-related problems, such as neoplasia, and cardiovascular and degenerative diseases [49]. In contrast, other studies in humans and seabirds found biphasic patterns of ROS associated with age, being higher in juvenile and aged individuals compared to those in the middle-aged groups [50,51]. Young animals need higher ROS to act as signal transducers for normal growth and development [50], while high ROS in older animals is linked to age-related diseases. Thus, oxidative stress responses associated with age can vary by individual depending on internal and external conditions [52], while patterns can also differ between short- and long-lived species [51]. There were no gender differences in ROS concentrations in this study, which contrasts with previous studies in humans that showed females usually have lower ROS than males due to higher concentrations of estrogenic hormones that have antioxidant properties and exposure to fewer external risk factors, such as smoking and alcohol [53,54]. Those external factors obviously do not apply to elephants, so the antioxidant functions of estrogens alone may not have enough of an effect to influence oxidative stress in this species. Lack of sex differences in ROS have also been found in other animal species, such as wild birds [55] and leopards [56], and between gelding and female horses [57]. Although there was no seasonal effect on ROS in the GLM model, there was an interaction between season and age, with ROS concentrations being lower in adult and aged groups during the winter. In rats [58] and cattle [59,60], the highest ROS concentrations were exhibited in summer, which partly agrees with our GLM pairwise comparison results of higher ROS in females in summer. In northern Thailand, weather differences across the three seasons are not extreme (THI range, 72.64 to 80.24 during our study), so it may not be surprising that there were minimal seasonal effects on oxidative stress markers. Animals living in other geographic regions with more variable THI may be expected to show more of a seasonal impact on ROS activity, as evidenced in cattle during summer in hotter areas of South Asia [47,60]. In addition, Maibam et al. [60] suggested that darker-pigmented skin in some cattle breeds may serve in a protective capacity to reduce oxidative stress due to heat stress. Elephant skin is fairly thick with moderate-to-dark-grey pigments, which may serve a similar protective purpose during the hotter summer months. However, there is no clear explanation as to why adult female elephants were more susceptible to increased ROS in summer, as the work activities were the same.

For MDA, there were no age or sex effects or interactions, which was comparable to human [61] and chimpanzee [62] data. In contrast, similar to ROS, other studies have proposed MDA may reach a peak at an older age based on sex differences in the elderly, who have lower estrogen concentrations [63,64]. Asian elephants do not appear to undergo hormonal menopause changes [65], so the lack of an age effect on oxidative stress markers might not be surprising. Based on chimpanzees, these markers (including MDA) only showed significant changes during ‘close-to-death’ events rather than just old age [62]. Our results indicate no seasonal differences in MDA concentrations in Asian elephants according to the GLM model, which contrasts with previous studies in cattle where the highest concentrations were exhibited in summer when high temperatures and relative humidity reduce the dispersal of body heat [47,66,67]. However, again, the THI in Thailand is relatively stable throughout the year, which could explain our lack of a strong seasonal effect.

For serum albumin, concentrations were highest in aged elephants (>45 years), which contrasts with humans that peak at around age 20 years of age [68] and, in fact, appear to decrease with increasing age [68,69]. In veterinary medicine, high serum albumin is generally linked to dehydration [70]. We did not assess hydration status or the amount of water consumed by elephants, but the use of aged elephants in tourist activities often is more limited, which means they are often restrained in the same place for prolonged periods [71] and so may have fewer opportunities to drink. The assessment of hydration status is complex, generally relying on multiple physiological and laboratory tests [72], but even then, high serum albumin alone is not enough to confirm dehydration. No statistical sex difference in serum albumin concentrations was found in our study, although in humans, it was generally higher in males but only at specific ages (20 to 60 years); concentrations then gradually normalize in both sexes by 60 years [68,73]. No seasonal effect on serum albumin in elephants was found, which agrees with some studies in humans [74,75] and cattle [76], although others have reported serum albumin does vary with the season, for example, being lower in winter in domestic dogs [77], higher in the rainy season in captive Ibex [78], and higher in winter in humans [79]. Seasonal differences in serum albumin are difficult to interpret because changing albumin concentrations can be the result of other factors, such as inflammation, nutrition changes, and fluid status [80]. The study in humans by Yanai et al. [79] presumed that a higher dietary intake during winter was linked to high serum albumin in this season, although there is no indication this is the case for captive elephants in Thailand. 

For GPx, there was no effect of age, which agrees with findings in other species [81,82,83]. In humans, GPx concentrations decrease with increasing age [61,84,85], especially after 65 years [86]. Longevity in elephants, which have the capacity to live more than 80 years, is not unlike that of humans [87]. In this study, only two elephants were over 60 years of age, so the population might not have been robust enough to identify a truly old-age effect. No sex difference in GPx was comparable to previous studies in humans [88] and dogs [89] but contrasts with a study in red deer that showed females had higher GPx activity than males [90], again perhaps because estrogens can facilitate modulation of various antioxidant enzymes, including GPx [91]. Seasonality did also not influence GPx activity, which contrasts with studies in deer and buffalos that reported lower GPx and other antioxidant markers in summer when compared to the winter season, possibly because these species are highly susceptible to heat stress [66,90].

For catalase, the lack of an age effect differed from previous studies in humans that showed an increase in catalase activity with increasing age [61,85]. However, Casado and Lόpez-Fernández [92] found only newborns (infants to 3 years) and the elderly (70–89 years) had significantly higher erythrocyte catalase activity than the other ages. No sex difference is in agreement with previous studies in humans [93,94], although Casado and Lόpez-Fernández [92] did find sex differences in an elderly group (more than 70 years) where the concentrations in males remained constant but were increased in older females. Although seasonality alone did not affect catalase activity based on the overall GLM model, pairwise comparisons showed that male (juvenile and subadult) and female (adult and aged) elephants in summer exhibited significantly lower catalase activity than in the other seasons, which is comparable to studies in cattle where lower levels of catalase were observed with higher ambient temperatures [66,95]. In humans, seasonal variation in catalase concentrations may be somewhat hormone dependent in that testosterone has pro-oxidant while estradiol has antioxidant properties, and melatonin has been shown to up-regulate a number of antioxidant enzymes [91,96].

For fGCMs, relationships with age are not always clear. In this study, there was no age effect, which was comparable to a study of semicaptive elephants in Myanmar [97]. However, a study of tourist camp elephants in Thailand found older animals exhibited overall lower fGCM concentrations [98]. One reason might be how age groups were defined, being across four broad age categories in this study and that of Seltmann et al. [97] compared to using age as a continuous variable in Bansiddhi et al. [98]. In terms of sex effects, those results have also not always been consistent, even in the same population. No sex difference agrees with data on semicaptive logging elephants in Myanmar (*n* = 75, 32 males, 43 females) [99]. However, another study of this same population with a larger sample size (*n* = 124, 49 males, 75 females) found males had higher fGCMs than females [97]. The current study had an even larger sample size (*n* = 137, 45 males, 92 females), yet no sex effect was evident. A number of extrinsic and intrinsic factors can affect fGCM concentrations in elephants [46,97,98,99,100,101,102,103], making it difficult to tease out demographic effects. It was surprising to see no seasonality effect on fGCMs in this study because numerous previous studies of captive elephants in Thailand showed that fGCM concentrations were higher in the winter season, which also corresponds to the high tourist season [46,100]. However, this study was conducted during the international travel ban in Thailand due to COVID-19 restrictions when all camps were closed [104], so this is the first evidence that seasonal changes in fGCMs may be directly related to tourist numbers and activities rather than environmental changes.

Regarding elephant health status, the results are preliminary as we only had a few animals within each category, with the greatest prevalence (*n* = 5 each) in the GI and EEHV groups. EEHV-HD elephants, by far, showed the most significant changes in a number of biological parameters, including those related to oxidative stress. Infected elephants exhibited the highest MDA concentrations, in addition to lower serum albumin and higher antioxidant enzymes (GPx and catalase) concentrations, suggesting this disease may be involved in oxidative stress responses. For MDA, high concentrations as a result of lipid peroxidation may indicate cell wall breakdown as a result of the increased production of free radicals from viral pathogenesis, which is a characteristic of herpesvirus infections in humans [105,106] and other species [11,12]. Another characteristic of EEHV-HD is systemic and lethal hemorrhaging [107,108]. Other hemorrhagic conditions, such as dengue fever in humans, have also been associated with an increase in MDA concentrations [37,39]. Significantly lower serum albumin in EEHV-HD cases may be related to severe endothelial cell damage that contributes to the leakage of protein molecules (i.e., albumin) into intercellular spaces, leading to facial edema [107]. Low levels of serum albumin and reductions in ROS-scavenging properties have been linked to poor cardiovascular disease outcomes in humans [30,109]; in elephants, EEHV-HD also affects cardiac function [110]. Unexpectedly, both enzymatic antioxidants (GPx and catalase) were noticeably higher during EEHV-HD, which differs from other studies that showed low levels of antioxidant enzymes associated with infectious diseases that cause hemorrhage [37,38,39], including other herpesvirus infections [12,106]. In these EEHV-HD cases, higher concentrations of oxidative markers were also not associated with lower concentrations of antioxidants as has been reported in other species [12,111], as normal physiological responses often involve the stimulation of antioxidants to counteract ROS production during the early stages of disease [112,113]. All EEHV-HD samples in this study were collected on the first day of clinical signs and prior to any treatments, so perhaps GPx and catalase had not had a chance to react to the oxidative challenge. Longitudinal studies are now needed to determine the time course of these changes and if they occur before clinical signs appear, something that could provide an early warning of pending poor outcomes. Finally, it is widely accepted that increased adrenal activity is associated with illness and injury in elephants [98,114]; however, no fGCM concentration differences were found in association with EEHV-HD in this study. In a previous study, one elephant calf exhibited a decrease in both salivary cortisol and fGCM concentrations 1 month prior to EEHV reactivation [101], so more longitudinal data leading up to clinical signs are needed, again to determine if such changes could be predictive of disease onset. Two calves survived, while the other three died not long after sample collection and treatment (e.g., fluid therapy, plasma infusion, and antiviral drugs) (<36 h). Although the sample size was small, Mann–Whitney U tests comparing survived and nonsurvived EEHV-HD found no significant differences in the studied biomarkers (Appendix A).

From the serum chemistry analyses, the average MCHC, an indicator of hemoglobin concentration in RBCs, was just below the reference range for EEHV-HD elephants, indicating a state of anemia. Thus, the MCHC may be the first red cell parameter to change in response to hemorrhage caused by EEHV-HD [107,110]. However, other red cell parameters were within the reference ranges. Heterophilia was found in the EEHV-HD group, while other types of white cells were within the normal range. This finding contrasts with a prior report that found a depletion of monocytes, lymphocytes, and heterophils in EEHV-HD calves [108], but this could simply reflect differences in time course changes relative to blood collection. An increase in heterophils is a response to infection or physiological stress [115,116]. Hence, the increased H:L ratio during EEHV-HD may be a sign of stress in infected calves [97,102], perhaps reflecting more acute changes than fGCMs, which, as a pooled value over time, did not identify a significant increase in these calves. Monitoring the M:H ratio has been suggested to be another indicator of clinical EEHV progression, with a decrease associated with disease [101], a finding confirmed in this study. Thrombocytopenia was found in the EEHV-HD group comparable to other studies [107,108]. High creatinine concentrations were also found in association with EEHV-HD, which agrees with previous reports [117,118], suggesting viral-induced renal lesions. ALT concentrations were 2 to 5 times higher in EEHV-HD-infected elephants, providing evidence of heart tissue damage [110]. Last, total serum protein concentrations were lower during EEHV-HD, again reflecting leakage of circulating proteins from endothelial damage due to the virus [107].

Not surprisingly, the responses of other health conditions were highly variable. There were no significant changes in oxidative markers or blood parameters in GI cases (*n* = 5), which was comparable to a report in horses that found unchanged oxidative stress markers in spasmodic colic [119], not unlike the condition in the study on elephants. However, in more severe GI cases in horses (e.g., colitis, obstructive colic), increased MDA and decreased catalase activity [119,120] have been observed, so these markers could be useful for determining the severity of GI cases in elephants. Of interest was that fGCM concentrations were above the reference range in two GI elephants (161.11 and 107.37 ng/g), which could indicate stress associated with abdominal pain as has been described in horses with colic [121,122].

Increased MDA concentrations are found in muscle injury and bone fracture conditions in humans and horses and associated with a build-up of free radicals from tissue damage and bone healing activity [123,124,125,126]. Similarly, in this study, elephants with MS problems (*n* = 4) had higher MDA concentrations; two had a broken bone, while the other two exhibited lameness from muscle pain. MDA concentrations in elephants with bone injuries were notably higher than in the muscle injury groups. Leg fractures in large animals, such as elephants, are critical because of problems with shifting body mass from the injured leg to other limbs that cause additional injury and lameness [6].

Overall, higher MDA concentrations in elephants with corneal problems (*n* = 3) were due to one elephant with a concentration of 4.33 nmol/mL. In that case, the eye condition had been ongoing for more than 2 weeks, whereas clinical signs in the other two were only evident for 1–2 days. Previous papers have reported an increase in local oxidative damage (including MDA) in corneal tissue and aqueous humor from corneal diseases in humans [127,128,129], so this biomarker could be informative, as eye problems are common in captive elephants [6,130].

There were no apparent changes in oxidative stress markers or blood parameters in the two elephants with puncture wounds that developed subcutaneous abscesses. In human studies, decreased albumin and total antioxidant capacities were found in wound fluids [131], while neopterin concentrations (an antioxidant marker) were over 100 times higher in abscess pus [132], suggesting oxidative stress pathways may be involved with wound processes. However, the use of serum in this study may not have adequately reflected the oxidative stress condition occurring within the local integument area.

The one elephant with chronic weakness had relatively low serum albumin (1.1 mg/dL). Serum albumin has been suggested as a mortality predictor, as low serum albumin usually contributes to poor treatment outcomes [109,133]. The concentration of fGCMs was also the highest in the study, over three times the high reference value, which agrees with previous findings of increased stress responses during disease events (e.g., GI problem, lameness, foot and skin lesions) in Asian elephants [98,103,114]. The H:L ratio has been suggested to be a stress indicator in Asian elephants [97,102], so the elevated H:L ratio further suggests this elephant was in a stressful state. Changes in three red cell parameters were also observed in this elephant, with lower PCV, hemoglobin, and RBC compared to the reference ranges, indicating anemia. These changes can occur in chronic disease conditions, such as infections, cancer, and kidney failure [134,135], although the cause of weakness in this elephant was not definitively diagnosed. Finally, ALT was increased by over 15 times the reference range, which indicates muscle tissue damage, no doubt due to prolonged recumbence (over 24 h) [136].

No changes in blood profiles were observed in sick elephants with GI, MS, eye, or wound problems. Previous case reports also showed no changes in hematology and serum chemistries in Asian elephants with GI issues [137], MS problems [138], or abscesses [139], although they have been noted in other species: colic in horses [140], lameness in goats [141], and abscesses in rats [142]. Hematology assessments are generally lacking in reports related to eye conditions (including corneal ulcers) in Asian elephants [130,143]; in horses with simple corneal ulcers, no changes in the hematological parameters were noted [144,145].

In constructing a correlation matrix for these biomarkers, a number of relationships were observed. Weak positive relationships between MDA and the antioxidants albumin and GPx were unexpected, considering oxidative stress pathways that suggest higher oxidative stress would relate to lower antioxidant levels [106,111]. However, mixed results in the associations between oxidative and antioxidative stress markers and not changing in predictive ways could be due to a number of factors that influence the oxidative status condition of individuals [12,111,146,147]. Again, captive environments in elephant camps (i.e., providing adequate housing and abundant diets) could mitigate oxidative stress impacts, as has been observed in a comparison between free-ranging and captive equids [12]. There was a positive association between catalase activity and the stress marker, fGCMs, which agrees with studies that showed increased catalase responses during physiological stress events [94,148], but it is not always consistent [149,150]. Few studies have been conducted on the association between blood parameters and oxidative stress markers. Our results show red blood cells positively correlated with oxidative status markers, comparable to previous studies in humans and dogs [111,151], potentially due to their physiological function of delivering oxygen to cells and providing an antioxidant system to counteract oxidative stress conditions [152].

## 5. Conclusions

This is the first study to measure multiple serum oxidative status markers in healthy and sick Asian elephants. Our results suggest that age and seasonal factors need to be considered when using these biomarkers for elephant health assessments. Longitudinal studies are still needed to determine annual changes in oxidative stress markers to see if there are any susceptible periods to oxidative stress in this species. If so, preventive strategies might be developed. Although there were limited samples from sick elephants, this study revealed significant responses for several biomarkers (significantly lower albumin and higher MDA, GPx, and catalase) in relation to EEHV-HD. Continued studies are now needed to monitor changes in these biomarkers, leading up to and continuing through the disease period, to better understand the pathophysiology and oxidative stress conditions in this and other health conditions.

## Figures and Tables

**Figure 1 animals-13-01548-f001:**
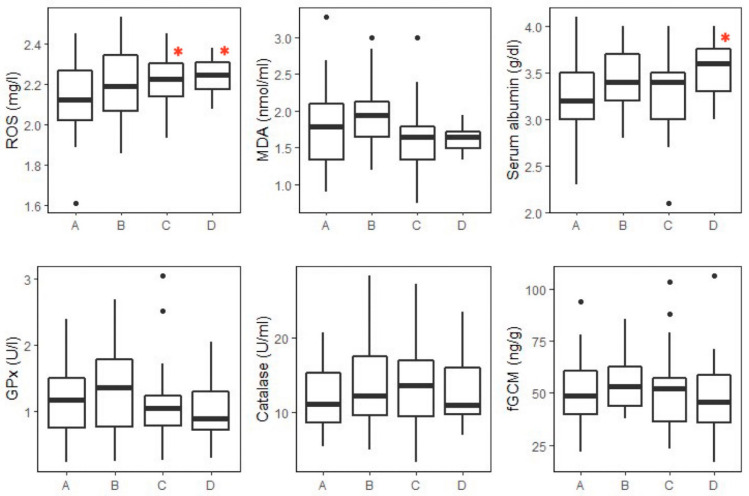
Box plots of oxidative status markers and fGCM concentrations across the four age groups of healthy elephants (*n* = 137): A = juvenile (1–5 years, *n* = 45); B = subadults (6–10 years, *n* = 36); C = adults (11–45 years, *n* = 45); and D = aged (>45 years, *n* = 11). Boxes indicate median, quartiles, and the 25th/75th percentiles. Whiskers represent the 10th/90th percentiles, and black dots are outliers. Asterisks indicate a significant difference at *p* < 0.05 according to the GLM model with juvenile as the reference. ROS: reactive oxygen species; MDA: malondialdehyde; GPx: glutathione peroxidase; fGCMs: fecal glucocorticoids.

**Figure 2 animals-13-01548-f002:**
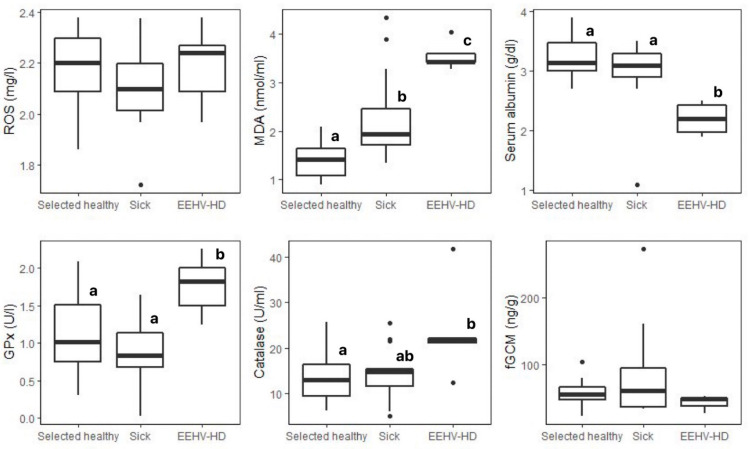
Box plots of oxidative status markers and fGCMs measured associated with health status of elephants: selected healthy (*n* = 30), sick (*n* = 15), and EEHV-HD (*n* = 5). Whiskers indicate median, quartiles, and the 25th/75th percentiles. Error bars represent the 10th/90th percentiles, and black dots indicate outliers. Different superscripts (a,b,c) indicate a significant difference between health groups within each biomarker at the *p* < 0.05 level according to Tukey post hoc tests. EEHV-HD: elephant endotheliotropic herpesvirus hemorrhagic disease; ROS: reactive oxygen species; MDA: malondialdehyde; GPx: glutathione peroxidase; fGCMs: fecal glucocorticoids.

**Table 1 animals-13-01548-t001:** General information on number of healthy elephants participating in this study, location of tourist camps, elephant daily activities, and main diet.

Area	Location	Number of Elephants	Number of Camps	Elephant Activities	Main Diet
North	Chiang Mai	104	20	Elephant trekking, observation, tourist feeding, bathing, playing in mud, elephant show	Fresh grasses, corn stalks
North	Lampang	16	1	Elephant trekking, observation, tourist feeding, bathing, elephant show	Fresh grasses, corn stalks
West	Kanchanaburi	4	2	Elephant trekking, tourist feeding, elephant show, bathing	Pineapple leaves, fresh grasses
Center	Phra Nakorn Si Ayutthaya	3	1	Elephant trekking, tourist feeding, elephant show	Pineapple leaves, fresh grasses
East	Surin	2	2	Tourist feeding, elephant show	Fresh grasses, corn stalks
East	Chonburi	8	1	Elephant trekking, tourist feeding, elephant show	Pineapple leaves, fresh grasses

**Table 2 animals-13-01548-t002:** Overall mean (±SD) and range of oxidative and stress markers in healthy elephants (*n* = 137) across age groups, sex, and sampling seasons. Number of animals assessed for each biomarker within each demographic group are shown in parentheses.

Biomarkers	Range	Overall	Age Group				Sex		Season		
			Juvenile 1–5 Years	Subadult 6–10 Years	Adult 11–45 Years	Aged >46 Years	Male	Female	Summer	Rainy	Winter
ROS (mg H_2_O_2_/L)	1.91–2.47	2.18 ± 0.16 (*n* = 137)	2.13 ± 0.17 (*n* = 45)	2.18 ± 0.17 (*n* = 36)	2.22 ± 0.13 (*n* = 45)	2.24 ± 0.09 (*n* = 11)	2.16 ± 0.17 (*n* = 45)	2.19 ± 0.15 (*n* = 92)	2.29 ± 0.13 (*n* = 33)	2.11 ± 0.17 (*n* = 53)	2.19 ± 0.11 (*n* = 51)
MDA (nmol/mL)	0.93–2.49	1.76 ± 0.47 (*n* = 137)	1.77 ± 0.49 (*n* = 45)	1.98 ± 0.48 (*n* = 36)	1.59 ± 0.43 (*n* = 45)	1.63 ± 0.19 (*n* = 11)	1.89 ± 0.51 (*n* = 45)	1.69 ± 0.45 (*n* = 92)	1.89 ± 0.35 (*n* = 33)	1.77 ± 0.49 (*n* = 53)	1.65 ± 0.51 (*n* = 51)
Serum albumin (g/dL)	2.73–3.96	3.33 ± 0.36 (*n* = 136)	3.22 ± 0.37 (*n* = 45)	3.45 ± 0.31 (*n* = 36)	3.29 ± 0.36 (*n* = 44)	3.54 ± 0.32 (*n* = 11)	3.40 ± 0.33 (*n* = 45)	3.29 ± 0.37 (*n* = 91)	3.35 ± 0.38 (*n* = 33)	3.32 ± 0.34 (*n* = 53)	3.32 ± 0.36 (*n* = 50)
GPx (U/L)	0.14–2.08	1.16 ± 0.56 (*n* = 136)	1.16 ± 0.53 (*n* = 45)	1.28 ± 0.62 (*n* = 36)	1.08 ± 0.53 (*n* = 44)	1.03 ± 0.53 (*n* = 11)	1.09 ± 0.51 (*n* = 45)	1.19 ± 0.58 (*n* = 91)	1.29 ± 0.56 (*n* = 33)	1.25 ± 0.57 (*n* = 53)	0.97 ± 0.49 (*n* = 50)
Catalase (U/mL)	2.99–21.86	13.05 ± 5.54 (*n* = 137)	11.71 ± 4.31 (*n* = 45)	13.68 ± 5.87 (*n* = 36)	13.95 ± 6.35 (*n* = 45)	12.72 ± 4.95 (*n* = 11)	13.3 ± 6.05 (*n* = 45)	12.9 ± 5.30 (*n* = 92)	8.96 ± 3.96 (*n* = 33)	12.9 ± 4.99 (*n* = 53)	15.80 ± 5.35 (*n* = 51)
fGCMs (ng/g feces)	22.23–82.70	51.97 ± 17.62 (*n* = 85)	52.13 ± 17.10 (*n* = 17)	55.59 ± 14.68 (*n* = 16)	51.12 ± 17.02 (*n* = 41)	49.61 ± 25.08 (*n* = 11)	51.67 ± 15.48 (*n* = 22)	52.07 ± 18.43 (*n* = 63)	44.58 ± 13.88 (*n* = 21)	58.08 ± 17.42 (*n* = 16)	53.16 ± 18.37 (*n* = 48)

ROS: reactive oxygen species; MDA: malondialdehyde; GPx: glutathione peroxidase; fGCMs: fecal glucocorticoids.

**Table 3 animals-13-01548-t003:** Mean (±SD) and ranges of biomarker concentrations in sick and elephant endotheliotropic herpesvirus hemorrhagic disease (EEHV-HD)-infected elephants compared to age- or sex- or camp-matched healthy elephants.

Biomarkers	Selected Healthy (*n* = 30) ^1^	Sick ^2^ (*n* = 15)	EEHV-HD (*n* = 5)	H Statistic	*p*-Value
ROS (mg/L)	2.18 ± 0.14 (1.86–2.38)	2.10 ± 0.16 (1.72–2.37)	2.19 ± 0.16 (1.97–2.38)	2.637	0.267
MDA (nmol/mL)	1.43 ± 0.35 ^a^ (0.90–2.49)	2.26 ± 0.91 ^b^ (1.34–4.33)	3.55 ± 0.29 ^c^ (3.28–4.03)	22.972	<0.01 **
Albumin (g/dL)	3.23 ± 0.33 ^a^ (2.70–3.90)	3.00 ± 0.62 ^a^ (1.10–3.50)	2.20 ± 0.29 ^b^ (1.90–2.50)	10.643	<0.01 **
GPx (U/L)	1.13 ± 0.47 ^a^ (0.30–2.09)	0.89 ± 0.39 ^a^ (0.30–1.64)	1.76 ± 0.40 ^b^ (1.24–2.25)	9.729	<0.01 **
Catalase (U/mL)	13.25 ± 4.85 ^a^ (6.33–25.64)	14.35 ± 5.56 ^ab^ (5.13–25.51)	23.81 ± 10.76 ^b^ (12.52–41.78)	6.083	0.048 *
fGCMs (ng/g)	55.43 ± 19.01 (22.15–103.23)	88.14 ± 77.05 (32.30–273.68)	42.00 ± 13.63 (26.48–52.03)	2.222	0.329

^a,b,c^ Different superscripts indicate a significant difference at *p* < 0.05. ^1^ Age-, sex-, and camp-matched. ^2^ Included weakness (*n* = 1), puncture wound (*n* = 2), gastrointestinal distress (*n* = 5), eye problems (*n* = 3), and musculoskeletal problems (*n* = 4). EEHV-HD: elephant endotheliotropic herpesvirus hemorrhagic disease; ROS: reactive oxygen species; MDA: malondialdehyde; GPx: glutathione peroxidase; fGCMs: fecal glucocorticoids. Asterisks indicated the significant levels at *p* < 0.05 (*) and *p* < 0.01 (**).

**Table 4 animals-13-01548-t004:** Mean (± SD) and range of blood parameters in healthy, sick, and EEHV-HD elephants.

Parameter		Unit	Range (Healthy)		Health Status	
				Healthy (*n* = 137)	Sick ^1^ (*n* = 15)	EEHV-HD (*n* = 5)
RBCs	PCV	%	27.62–47.52	37.51 ± 5.97	32.82 ± 2.29	35.80 ± 7.56
	Hemoglobin	g/dL	10.29–16.10	13.31 ± 1.89	11.87 ± 0.78	12.34 ± 3.06
	RBC count	×10^6^ cells/µL	2.39–3.91	3.15 ± 0.45	2.70 ± 0.20	3.24 ± 0.65
	MCV	fl	106.30–129.08	117.72 ± 6.68	121.85 ± 3.65	110.98 ± 3.43
	MCHCs	g/dL	34.62–36.82	35.80 ± 0.73	36.16 ± 0.89	34.20 ± 2.83
WBCs	WBC count	cells/µL	7130.58–20,368.19	13,972.19 ± 4040.26	12,467.50 ± 4457.69	18,950.00 ± 11,131.23
	Heterophil	cells/µL	974.61–5383.52	3506.49 ± 1596.61	3672.50 ± 1942.69	9502.80 ± 4755.95
	Lymphocyte	cells/µL	686.23–9985.99	5529.07 ± 2590.80	4029.42 ± 1642.62	5909.40 ± 2778.33
	Monocyte	cells/µL	422.91–7459.65	4173.14 ± 2066.36	4112.42 ± 1833.32	3351.40 ± 4078.38
	Eosinophil	cells/µL	0–598.60	268.08 ± 267.25	626.75 ± 1331.35	186.40 ± 215.53
	Basophil	cells/µL	0–100.33	33.10 ± 62.36	24.83 ± 86.02	Not found
	H:L ratio		>0–1.47	0.79 ± 0.63	1.07 ± 0.63	1.59 ± 0.26
	M:H ratio		>0–2.73	1.39 ± 0.88	1.31 ± 0.64	0.30 ± 0.24
Platelets	Platelet count	×10^3^ cells/µL	151.08–552.62	361.38 ± 120.19	400.25 ± 65.50	118.40 ± 103.71
Blood chemistry	BUN	mg/dL	4.44–15.31	10.25 ± 3.42	8.67 ± 2.80	12.80 ± 2.00
	Creatinine	mg/dL	0.90–1.89	1.42 ± 0.29	1.34 ± 0.39	2.20 ± 0.33
	ALT	U/L	0–2.11	0.92 ± 1.03	4.15 ± 9.92	5.75 ± 3.59
	ALP	U/L	0–306.92	154.59 ± 93.68	100.80 ± 73.26	113.00 ± 28.08
	Total serum protein	g/dL	6.81–9.91	8.33 ± 0.85	8.10 ± 0.82	5.57 ± 0.76

EEHV-HD: elephant endotheliotropic herpesvirus hemorrhagic disease; PCV: packed cell volume; RBCs: red blood cells; MCV: mean corpuscular volume; MCHCs: mean corpuscular hemoglobin concentrations; WBCs: total white blood cells; H:L ratio: heterophil-to-lymphocyte ratio; M:H ratio: monocyte-to-heterophil ratio; BUN: blood urea nitrogen; ALT: alanine transaminase; ALP: alanine phosphatase. ^1^ Included weakness (*n* = 1), puncture wound (*n* = 2), gastrointestinal distress (*n* = 5), eye problems (*n* = 3), and musculoskeletal problems (*n* = 4).

**Table 5 animals-13-01548-t005:** Correlation matrix for oxidative status markers (ROS, MDA, albumin, GPx, and catalase), fGCMs, and different blood cell parameters in healthy elephants (*n* = 137).

Parameters	ROS	MDA	Albumin	GPx	Catalase	fGCMs	RBCs	WBCs	HETs	LYMs	MONOs	PLTs
ROS												
MDA	−0.024											
Albumin	0.09	0.26 **										
GPx	0.15	0.29 **	0.02									
Catalase	−0.09	0.15	0.18	0.17								
fGCMs	−0.36 **	−0.15	−0.08	−0.08	0.23 *							
RBCs	0.27 **	0.27 **	0.23 **	0.22 *	0.18 *	−0.33 **						
WBCs	−0.01	0.06	−0.03	0.16	0.11	0.04	−0.01					
HETs	−0.11	−0.14	−0.18 *	0.05	0.01	0.17	−0.25 **	0.57 **				
LYMs	0.11	0.30 **	0.19 *	0.10	0.17	−0.11	0.19 *	0.58 **	0.01			
MONOs	0.10	−0.07	−0.07	0.18 *	0.07	−0.21	0.04	0.46 **	0.05	−0.06		
PLTs	−0.25 **	−0.10	−0.18 *	0.10	−0.04	0.30 **	−0.23 **	0.63 **	0.52 **	0.11	0.44 **	

ROS: reactive oxygen species; MDA: malondialdehyde; GPx: glutathione peroxidase; fGCMs: fecal glucocorticoids; RBCs: red blood cells; WBCs: total white blood cells; HETs: heterophils; LYMs: lymphocytes; MONOs: monocytes; PLTs: platelets. Asterisks indicated the significant levels at *p* < 0.05 (*) and *p* < 0.01 (**).

## Data Availability

The data presented in this study are available on request from corresponding authors.

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
