# Peer review of "Measures of Oxidative Status Markers in Relation to Age, Sex, and Season in Sick and Healthy Captive Asian Elephants in Thailand"

_animals, 2023, doi:10.3390/ani13091548_

Round 1

Reviewer 1 Report

This manuscript documents oxidative status markers in Asian elephants across age, sex, season, and health status. The article is well written, and the results included provide important reference values for future studies. Results described for EEHV animals are very interesting and could be useful in the future as biomarkers of disease status.

A few minor suggestions:

1. A subset of the healthy elephants were selected to compared to the sick animals. It will be helpful to see which animals specifically were selected instead of simply stating that they were age and sex matched. This age/sex matching approach should be shown in a table.

2. In figure 2, a, b, c, and ab are used to indicate significant values, but the legend says that they are all p<0.05. Why are different letters needed to indicate this?

3. Because the EEHV-HD data showed the most changes in oxidative status markers, it would be helpful to know if these animals survived or succumbed to disease. It would also be helpful to know if they were being treated at the time of sample collection.

4. Line 496 of the discussion states that elephants "can live more that 80 years, is second only to humans." This statement is not accurate. There are animals that live longer than humans and elephants, whales and tortoises are just a couple of examples.

5. Can you add a reference stating that a 12 hour window is feasible for measuring CBC in elephants? Some blood cells are short lived and die within hours.

Author Response

Response to Reviewer 1 Comments

Overall comments

This manuscript documents oxidative status markers in Asian elephants across age, sex, season, and health status. The article is well written, and the results included provide important reference values for future studies. Results described for EEHV animals are very interesting and could be useful in the future as biomarkers of disease status.

Response:

Thank you very much for your supportive comments.

Point 1: A subset of the healthy elephants were selected to compared to the sick animals. It will be helpful to see which animals specifically were selected instead of simply stating that they were age and sex matched. This age/sex matching approach should be shown in a table.

Response 1:

Thank you for your suggestion. We have now added a supplementary table (Table S1) for individual healthy elephant data that were selected according to age and sex matching to sick elephants in the study. Note we do not specify the camp over privacy concerns.

We have added sentences in the manuscript in Line 263-264; “Data on age and sex of sick and selected healthy elephants are shown in Table S1.”

Table S1. Data on age and sex of sick and selected healthy elephants in this study.

Sick elephants

(N=20)

Selected healthy (N=30)

Age / sex

Conditions

Age / sex

EEHV-HD

1y / male

2y / male

EEHV-HD

2y / male

2y / male

EEHV-HD

4y / male

2y / male

3y / male

3y / male

4y / male

EEHV-HD

3y / female

3y / female

EEHV-HD

4y / female

4y / female

MS

5y / female

4y / female

4y / female

5y / female

5y / female

Eye

11y / female

9y / female

10y / female

MS

14y / female

15y / female

GI

24y / female

23y / female

MS

31y / female

30y / female

GI

35y / female

33y / female

Wound

35y / female

37y / female

GI

40y / female

40y / female

41y / female

MS

42y / female

42y / female

42y / female

Eye

47y / female

46y / female

46y / female

Eye

48y / female

48y / female

GI

50y / female

50y / female

GI

50y / female

53y / female

Wound

50y / female

54y / female

Weakness

55y / female

60y / female

Point 2: In figure 2, a, b, c, and ab are used to indicate significant values, but the legend says that they are all p<0.05. Why are different letters needed to indicate this?

Response 2:

Different superscripts in a figure are shown to indicate differences between groups for the same biomarkers, which ones were significant at the P<0.05 level. Not all biomarkers showed health group differences and so they did not have superscripts. We have modified the legend to try and make this more clear “Different superscripts (a,b,c) indicate a significant difference between health groups within each biomarker at the P<0.05 level according to the Kruskal-Wallis tests.”.

Point 3: Because the EEHV-HD data showed the most changes in oxidative status markers, it would be helpful to know if these animals survived or succumbed to disease. It would also be helpful to know if they were being treated at the time of sample collection.

Response 3:

Thank you for your suggestion. We have added details of EEHV-infected elephants regarding treatment outcome in the manuscript as follows:

Lines 578-581; “Two calves survived while the other three died not long after sample collection and treatment (e.g., fluid therapy, plasma infusion, and antiviral drug) (<36 h). Although the sample size was small, Mann-Whitney U tests compared between survived and non-survived EEHV-HD found no significant differences in studied biomarkers (Table S11).”

Table S11. Biomarker concentrations (mean ± SD) and statistical comparisons between calves that survived or succumbed to EEHV-HD using the Mann-Whitney U tests.

Biomarkers

Survived

(N=2)

Died

(N=3)

U statistic

P-value

ROS (mg/l)

2.31 ± 0.09

2.11 ± 0.15

1

0.386

MDA (nmol/ml)

3.36 ± 0.11

3.68 ± 0.31

5.5

0.236

Albumin (g/dl)

2.25 ± 0.35

2.15 ± 0.35

1

0.698

GPx (U/l)

1.53 ± 0.41

1.92 ± 0.39

5

0.386

Catalase (U/ml)

17.36 ± 6.85

28.29 ± 11.69

4

0.773

fGCM (ng/g)

52.03 (N=1)

36.98 ± 14.86

0

0.540

Point 4: Line 496 of the discussion states that elephants "can live more that 80 years, is second only to humans." This statement is not accurate. There are animals that live longer than humans and elephants, whales and tortoises are just a couple of examples.

Response 4:

Thank you for your comments and you are correct! There are many other species that live longer than elephants and humans. We have edited the manuscript accordingly.

Line 497: “Longevity in elephants, which have the capacity to live more than 80 years, is not unlike that of humans [87].

Point 5: Can you add a reference stating that a 12 hour window is feasible for measuring CBC in elephants? Some blood cells are short lived and die within hours.

Response 5:

Interesting question. However, to our knowledge, there is no study that has documented hematology changes and EDTA storage over time in Asian elephants. The previous hematology study in captive Asian elephants was processing CBC and serum chemistry within 24 h of sample collection (Janyamethakul et al., 2017). The previous study in humans suggested that 4OC is recommended as a suitable storage temperature for 12 h storage (Unalli and Ozarda, 2021).

We have edited the manuscript to try and clarify this more.

Line 173-177; “EDTA blood was chilled (4OC) and used for complete blood counts conducted immediately after transportation to FVM-CMU (within 6-12 h) by auto hematology analyzer (BC-5300, MindrayTM, Shenzhen, P.R.China), whereas white blood cell differential counts were done manually on Wright-Giemsa-stained blood smears, similar to sample processing protocols developed for humans [42].”

References

Janyamethakul, T., Sripiboon, S., Somgird, C., Pongsopawijit, P., Panyapornwithaya, V., Klinhom, S., Loythong, J., Thitaram, C., 2017. Hematologic and biochemical reference intervals for captive Asian elephants (Elephas maximus) in Thailand. Kafkas. Univ. Vet. Fak. Derg. doi:10.9775/kvfd.2017.17380

Unalli, O., Ozarda, Y., 2021. Stability of hematological analytes during 48 hours storage at three temperatures using Cell-Dyn hematology analyzer. J Med Biochemistry 40, 252–260. doi:10.5937/jomb0-27945

Reviewer 2 Report

Authors present an important work regarding the study of multiple biomarkers changes in Asian elephants. I consider the value and novelty of this work. I think the goals and the idea are very relevant. However, there are some aspects that concerned me while reading your work. The main issue that is related to all the following concerns is the small number of assessed individuals, even though I totally understand that with some species (especially the endangered ones) is very hard to correct this problem. Please find my comments on your manuscript above. 

1) I do not clearly understand the potential of measuring oxidative stress biomarkers in a clinical diagnosis of a single elephant patient. As you said, alterations are very unspecific and due to physiological or pathological conditions. I do understand its use in figuring out some population health hazards (e.g the effects of chemical pollution in some organs, as the liver or kidney). I would understand if you want to study a specific health condition (as EEHV), compared to healthy animals. However, grouping all the sick conditions into one group seem to vague for me.

2) On the other hand, I was also confused when you divide your elephants in three groups (health, sick and EEHV). Why is EEHV a separate group and what is the interest in separating a disease from all the others and mixing, for instance, GI disease with eye problems, in the first tables? What practical conclusions can be taken from this? 

3) You have very few animals to compare each condition as you do in the second part of the Discussion, and it is almost impossible to take most of the conclusions you take, in my opinion. Have you considered simplifying your study (e.g number of biomarkers, variants...) and present your results as a Short Communication? You call your results preliminary (line 540), so maybe it makes sense. I understand there is a lot of information presented to a short communication, but actually, I think most arguments and conclusions you discuss in your manuscript can not be well supported due to the number of animals in each group or health condition.

Other aspects:

Line 70- "Similar to other species" (not only wild animals)

Line 265 - please indicate in this section the critical value of p that you consider in your statistical tests.

Thank you for the opportunity to revise this paper. I wish the authors a good work.

Author Response

Response to Reviewer 2 Comments

Overall comments

Authors present an important work regarding the study of multiple biomarkers changes in Asian elephants. I consider the value and novelty of this work. I think the goals and the idea are very relevant. However, there are some aspects that concerned me while reading your work. The main issue that is related to all the following concerns is the small number of assessed individuals, even though I totally understand that with some species (especially the endangered ones) is very hard to correct this problem.

Response:

Thank you very much for your comments and understanding of our limitations in working with endangered species like Asian elephants. We did consider the number of animals for establishing the reference values for these biomarkers. According to protocols by the American Society of Veterinary Clinical Pathology, a minimum of 120 animals are needed for establishing reference intervals in an animal species (Friedrichs et al., 2012). Hence, we have passed the minimum requirement with this number of healthy animals (N=132). However, we recognize that we have only a small number of ‘sick’ elephants that we wanted to compare with healthy animals. The main goal was to start building a database of biomarker responses to those conditions and provide directions for future studies related to oxidative stress conditions in this species. We believe we have accomplished this goal, but have edited the text to make sure the preliminary nature of these data is clear.

References

Friedrichs, K.R., Harr, K.E., Freeman, K.P., Szladovits, B., Walton, R.M., Barnhart, K.F., Blanco-Chavez, J., 2012. ASVCP reference interval guidelines: determination of de novo reference intervals in veterinary species and other related topics. Vet Clin Pathol 41, 441–453. doi:10.1111/vcp.12006

Point 1: I do not clearly understand the potential of measuring oxidative stress biomarkers in a clinical diagnosis of a single elephant patient. As you said, alterations are very unspecific and due to physiological or pathological conditions. I do understand its use in figuring out some population health hazards (e.g the effects of chemical pollution in some organs, as the liver or kidney). I would understand if you want to study a specific health condition (as EEHV), compared to healthy animals. However, grouping all the sick conditions into one group seem to vague for me.

Response 1:

Thank you for your comments. Not only hazardous events (e.g., pollution, environmental changes), but many studies in human medicine have also been studied a change of oxidative stress markers in various disease conditions, for example, cardiovascular diseases (Ho et al., 2013), diabetes mellitus (Dworzański et al., 2020), herpesvirus infection (Jakovljevic et al., 2018), as well as other species such as colic in horses (Ibrahim, 2014), parasitic infestation in goats and sheep (Alam et al., 2020). Hence, we believed that we might expect some specific changes with regards to aspects of elephant disease as well.

As you point out, we have small number in each of the ‘sick’ categories, which was not enough to run robust statistical analysis on all sick groups. But again, the main goal was to create a database of biomarker responses to various conditions that might aid in better understading health risks and provide directions for future studies related to oxidative stress conditions in elephants. We believe it is important to publish even preliminary data that can be used by other researchers interested in this area, and who can build on this work. Hence, in the manuscript, we decided to group ‘sick’ elephants into one category and compare them to a disease that is a major concern worldwide (i.e., EEHV-HD), a fatal viral disease in elephant calves and heavily impact on elephant conservation worldwide. This disease also exhibited the most biomarker changes. For other diseases, we show biomarker and hematology data separated in each ‘sick’ elephant in Table S3 and S10 (the new supplementary file). We also discuss each state and point out the interesting responses we found. We hope the data are informative enough for readers to follow and get an idea what conditions may or may not cause a change of studied biomarkers for future study.

References

Alam, R.T.M., Hassanen, E.A.A., El-Mandrawy, S.A.M., 2020. Heamonchus Contortus infection in Sheep and Goats: alterations in haematological, biochemical, immunological, trace element and oxidative stress markers. Journal of Applied Animal Research 48, 357–364. doi:10.1080/09712119.2020.1802281

Dworzański, J., Strycharz-Dudziak, M., Kliszczewska, E., Kiełczykowska, M., Dworzańska, A., Drop, B., Polz-Dacewicz, M., 2020. Glutathione peroxidase (GPx) and superoxide dismutase (SOD) activity in patients with diabetes mellitus type 2 infected with Epstein-Barr virus. PLOS ONE 15, e0230374. doi:10.1371/journal.pone.0230374

Ho, E., Karimi Galougahi, K., Liu, C.-C., Bhindi, R., Figtree, G.A., 2013. Biological markers of oxidative stress: applications to cardiovascular research and practice. Redox Biol. 1, 483–491. doi:10.1016/j.redox.2013.07.006

Ibrahim, H.M.M., 2014. Oxidative stress associated with spasmodic, flatulent, and impaction colic in draft horses. J. Equine Vet. Sci. 34, 1205–1210. doi:10.1016/j.jevs.2014.08.002

Jakovljevic, A., Andric, M., Knezevic, A., Miletic, M., Beljic-Ivanovic, K., Milasin, J., Sabeti, M., 2018. Herpesviral infection in periapical periodontitis. Curr. Oral Health Rep. 5, 255–263. doi:10.1007/s40496-018-0198-7

Point 2: On the other hand, I was also confused when you divide your elephants in three groups (health, sick and EEHV). Why is EEHV a separate group and what is the interest in separating a disease from all the others and mixing, for instance, GI disease with eye problems, in the first tables? What practical conclusions can be taken from this?

Response 2:

We agree we had a limited number of sick elephants, with some groups that had low elephant numbers that were not statistically testable. Hence, we decided to group all sick elephants into one group and separate out EEHV-HD, which did have a testable number of cases (N=5) because it is the most severe viral disease in elephant calves and can be fatality when compared to other disease conditions in this study. We believe that it is important to show how EEHV-HD affected these biomarkers specifically, as this disease is one of the most critical in elephant medicine, and more research is needed to improve diagnosis, treatment, and pathophysiology. Our data suggests some oxidative biomarkers may be informative in EEHV cases.

            But we also felt it was important to show biomarker concentrations and hematology values in each disease condition (Table S3 and S10 and new supplementary file). Again, to highlight how these conditions might alter stress biomarkers, which can provide ideas for future study.

Point 3: You have very few animals to compare each condition as you do in the second part of the Discussion, and it is almost impossible to take most of the conclusions you take, in my opinion. Have you considered simplifying your study (e.g number of biomarkers, variants...) and present your results as a Short Communication? You call your results preliminary (line 540), so maybe it makes sense. I understand there is a lot of information presented to a short communication, but actually, I think most arguments and conclusions you discuss in your manuscript can not be well supported due to the number of animals in each group or health condition.

Response 3:

Thank you for your suggestion. We have tried our best to not overinterpretation the data with the small number of animals in the sick categories, but as described above, we do think presenting these data in total is important and a way to point to the need for additional studies to confirm or refute these findings. We also believe a full paper is needed because this is the first study to investigate these serum oxidative stress markers in the context of various disease states in this species. Our intention is to continue collecting samples from elephants to further characterize these responses, but also to provide data that other researchers could potentially build upon given some of the interesting results we found.

Point 4: Line 70- "Similar to other species" (not only wild animals)

Response 4:

Thank you for your suggestion. We have edited the sentence following your comments.

Line 70; “Similar to other species”

Point 5: Line 265 - please indicate in this section the critical value of p that you consider in your statistical tests.

Response 5:

Thank you for your suggestion. We have now added “Statistical significance for all tests was set at P<0.05.” in Line 268.

Thank you for the opportunity to revise this paper. I wish the authors a good work.

Response:

Thank you very much for your valuable comments. We have tried to carefully respond to all the comments. We hope the newly edited version is appropriate for publication.

Round 2

Reviewer 2 Report

The authors have answered all my reviewer comments. Aalthough most of the answers are justifications for performing the study this way, they are well-argued.

I did not say in my review report that oxidative stress biomarkers are only significant as effects of pollution, I was just giving an example. My point is: they are not very specific to be analysed with a very diverse group of sick animals with different conditions. The real information that can be extracted is reduced.

I sincerely agree that these data should be published and I support the publication. But not in the current form. I still do not agree that it should be analysed by grouping all the sick elephants (even for statistical reasons).  If your focus is the biomarkers' changes in EEHV-ED elephants I believe you should compare those with normal elephants. I still do not agree with the three groups as an option.

But I will not insist more, I already express my opinion. I honestly recognize the value of your data, that is not a question at all. My issue is with the interpretation, the way you analysed them and, consequently, the presentation. 

I wish you all the best
